# How should COVID-19 vaccines be distributed between the Global North and South: a discrete choice experiment in six European countries

Janina I Steinert[1]*[†], Henrike Sternberg[2†], Giuseppe A Veltri[3], Tim Büthe[4]

[1]Hochschule für Politik/Munich School of Politics and Public Policy, TUM School of Social Sciences & Technology and TUM School of Medicine, Technical University of Munich, Munich, Germany; [2]TUM School of Social Sciences & Technology, Technical University of Munich, Munich, Germany; [3]Department of Sociology and Social Research, University of Trento, Trento, Italy; [4]Hochschule für Politik/Munich School of Politics and Public Policy and TUM School of Social Sciences & Technology, Munich, Germany

## Abstract

**Background:** The global distribution of COVID-19 vaccinations remains highly unequal. We examine public preferences in six European countries regarding the allocation of COVID-19 vaccines between the Global South and Global North.

**Methods:** We conducted online discrete choice experiments with adult participants in France (n=766), Germany (n=1964), Italy (n=767), Poland (n=670), Spain (n=925), and Sweden (n=938). Respondents were asked to decide which one of two candidates should receive the vaccine first. The candidates varied on four attributes: age, mortality risk, employment, and living in a low- or high-income country. We analysed the relevance of each attribute in allocation decisions using conditional logit regressions.

**Results:** In all six countries, respondents prioritised candidates with a high mortality and infection risk, irrespective of whether the candidate lived in the respondent's own country. All else equal, respondents in Italy, France, Spain, and Sweden gave priority to a candidate from a low-income country, whereas German respondents were significantly more likely to choose the candidate from their own country. Female, younger, and more educated respondents were more favourable to an equitable vaccine distribution.

**Conclusions:** Given these preferences for global solidarity, European governments should promote vaccine transfers to poorer world regions.

**Funding:** Funding was provided by the European Union's Horizon H2020 research and innovation programme under grant agreement 101016233 (PERISCOPE).

**\*For correspondence:**
janina.steinert@tum.de

[†]These authors contributed equally to this work

**Competing interest:** The authors declare that no competing interests exist.

## Editor's evaluation

Despite all efforts, the global distribution, developed versus developing economies, of Covid-19 vaccines and immunisation with them remains highly unequal. This rigorous and well-designed study addresses the question of what are the public preferences for vaccine allocation globally in 6 European countries. The results documented in this paper show that overall the public preference is for equitable distribution and provide important evidence and insights for policymakers.

## Introduction

In his opening speech to address the United Nations General Assembly in September 2021, Secretary General António Guterres expressed stark discontent with the highly unequal global distribution of COVID-19 vaccines: *"A majority of the wealthier world is vaccinated. Over 90 percent of Africans are still waiting for their first dose. This is a moral indictment of the state of our world. It is an obscenity"* (**United Nations Secretary-General, 2021**). At the time of writing, and a year after Guterres gave his speech, 80% of the citizens of low-income countries are still waiting for their first dose of a COVID-19 vaccine, whereas in high-income countries, almost 80% of citizens are vaccinated. While the initial sense of urgency at the onset of the pandemic may have passed, this blatant inequity in access to COVID-19 vaccines prevails, continuing also throughout the more recent booster campaigns: In the past 12 months, the total number of vaccine doses administered per 100 people amounts to almost 100 in high-income countries and to only 27 in low-income countries (**Mathieu et al., 2021**; **Our World in Data, 2022**). It therefore also seems likely that we will observe this same pattern again with the new Omicron-specific vaccine.

Achieving high COVID-19 vaccination rates globally is imperative for three reasons. From an ethical perspective, unequal access to vaccination leads to high rates of transmission, severe infections, and deaths in those parts of the world where health care capacity is the lowest. This aggravates existing health inequities between the Global South and North (**Godlee, 2021**; **Katz et al., 2021**; **Krause et al., 2021**; **Wagner et al., 2021**). In addition, achieving a more equitable vaccine distribution would have utilitarian benefits: A recent modeling study compared two hypothetical scenarios – one in which the 50 richest countries used all available vaccines and one in which vaccines were allocated to all countries proportionally to their population size – finding that the former scenario would lead to twice as many COVID-19 deaths (**Herzog et al., 2021**). Second, there are compelling economic arguments in favour of equitable vaccine distribution: The RAND Corporation estimates that constrained access to COVID-19 vaccines in low- and middle-income countries (LMICs) would reduce the global GDP by US$ 153 billion each year, including a loss of US$ 56 billion in the European Union and United States combined (**Hafner et al., 2020**). Put differently, every US dollar spent on supplying vaccines to LMICs would yield a return of US$ 4.8 (**Hafner et al., 2020**). Third, alleviating global asymmetries in COVID-19 vaccine coverage is warranted for virologic reasons. Unmitigated COVID-19 transmissions in some parts of the world create evolutionary reservoirs from which new SARS-CoV-2 variants arise, increasing the risk of immune escape - for both vaccine-induced and natural immunity - and of other phenotypic changes that could lead to greater virulence (**Saad-Roy et al., 2021**; **Telenti et al., 2021**; **van Oosterhout et al., 2021**; **Wagner et al., 2021**). As WHO Director General Tedros Adhanom Ghebreyesus put it: "none of us will be safe until everyone is safe" (**WHO Director General, 2022**).

The globally unequal distribution of COVID-19 vaccines, including the stockpiling of vaccine doses for their own citizens, is partly a consequence of widespread vaccine nationalism in high-income countries (**Harman et al., 2021**; **Herzog et al., 2021**; **Wagner et al., 2021**). To ensure 'fair and equitable access' to COVID-19 vaccines for all countries and achieve high vaccination rates everywhere, the WHO, the Vaccine Alliance (Gavi), and the Coalition for Epidemic Preparedness Innovations (CEPI) formed a multilateral initiative named 'COVID-19 Vaccines Global Access', COVAX (**Herzog et al., 2021**). However, several governments have resorted to making bilateral purchasing agreements with vaccine manufacturers outside of COVAX, which has substantially weakened the initiatives' collective purchasing power (**Kim, 2021**; **Wouters et al., 2021**). Moreover, out of those vaccine doses that were initially announced as donations to COVAX by high-income countries, substantial proportions – 25% of announced EU doses and almost 50% of announced US doses – have in fact not yet been donated (WHO ACT-Accelerator Hub; **Our World in Data, 2022**; **Mathieu et al., 2021**). The WHO's pledges for a more equitable COVID-19 vaccine distribution have thus still not been fulfilled.

Decision-makers in high-income countries are likely to exhibit vaccine nationalism and refrain from vaccine donations if they believe there is little public support for giving COVID-19 vaccines to poorer regions of the world (**Clarke et al., 2021**). Thus, governments will likely only donate vaccines or actively participate in international vaccine alliances such as COVAX if they do not expect to pay a price at the ballot box. A thorough understanding of public preferences for the global distribution of COVID-19 vaccines is therefore paramount.

Recent empirical literature has explored public preferences for the allocation of COVID-19 vaccines. A majority of these studies examined public opinion on *prioritisation within* high-income countries and

when COVID-19 vaccine availability was still heavily constrained in those countries (*Duch et al., 2021*; *Gollust et al., 2020*; *Knotz et al., 2021*; *Luyten et al., 2020*; *Persad et al., 2021*; *Reeskens et al., 2021*; *Sprengholz et al., 2021*). Based on data from online surveys and survey experiments, the studies revealed substantial public support for prioritising frontline healthcare workers and clinically vulnerable groups (*Duch et al., 2021*; *Persad et al., 2021*).

To our knowledge, only four studies to date have examined individuals' preferences on the distribution of COVID-19 vaccines across national borders (*Clarke et al., 2021*; *Guidry et al., 2021*; *Klumpp et al., 2021*; *Vanhuysse et al., 2021*). One online survey conducted in seven high-income countries found that around 50% of participants generally supported global allocation schemes that would give priority to the countries that could not afford to purchase vaccines (*Clarke et al., 2021*). Another survey conducted in the US found that 40% of respondents were in favour of donating at least 10% of the nationally purchased vaccines to poorer countries. Support was less pronounced among older respondents – a group that is at greater risk of severe disease progression if infected (*Guidry et al., 2021*).

A survey conducted in Germany asked participants to choose between different options for international agreements and alliances on the distribution of COVID-19 vaccines, which varied by (i) countries joining the agreement, (ii) distribution rules, and (iii) cost per German household. The authors found that participants displayed a strong preference for an alliance exclusively composed of EU states. More importantly, the authors found that participants were more supportive of vaccine alliances if the national cost of participation was lower and national vaccine coverage higher, suggesting that participants' preferences were significantly shaped by self-interest (*Vanhuysse et al., 2021*).

In contrast, in a discrete choice experiment conducted in Germany and the US, participants in both countries expressed a strong preference for prioritising vaccine allocation to countries with a higher number of COVID-19 deaths and fewer intensive care unit beds, even when they were asked to imagine that they or a vulnerable family member were still waiting for the COVID-19 vaccine (*Klumpp et al., 2021*). Notably, no previous study to date has exclusively sampled participants who were still waiting for their first COVID-19 vaccine dose when participating in the survey experiment.

In this paper, we analyse new experimental evidence from six EU countries on citizens' preferences for the distribution of COVID-19 vaccines between the Global South and North. We advance the literature in three ways. First, by covering six countries, we implement the largest survey experiment on international vaccine allocation preferences to date and are thus able to examine differences in citizens' preferences across EU member states. Second, we conduct a discrete choice experiment among participants who are themselves not yet vaccinated, asking them to allocate a COVID-19 vaccine to either a person in their own country or to a person in a country in the Global South. This places specific salience on the notion that donating a vaccine dose to a person in the Global South might mean sacrificing one's own dose or that of a fellow citizen, thus leveraging self-interest-based and nationalistic considerations. Third, we specifically examine heterogeneity in participants' preferences along key sociodemographic characteristics as well as in terms of regional COVID-19 case numbers. Policymakers can use these insights to anticipate which groups will be most supportive of vaccine donations and might thus be at the heart of a supporting coalition - and to which groups political leaders, epidemiological experts, or civil society groups might still need to provide better information about the benefits for the donating country.

## Materials and methods
### Study sample
We conducted an online survey experiment in six EU countries: France, Germany, Italy, Poland, Spain, and Sweden. The German survey was launched at the height of the third wave in April 2021; the other five surveys were carried out in June 2021, coinciding with a phase of low case numbers in each country (see *Figure 1*). In each country, we recruited respondents aged 18 years and older, drawing on online panels of the survey provider *Bilendi-Respondi*. We sampled participants based on quotas that were matched to the census population of each target country in terms of (1) gender, (2) age, (3) education, and (4) geographic location (e.g. state or province within each country) (see *Supplementary file 1a* for the census statistics of the sampled countries).

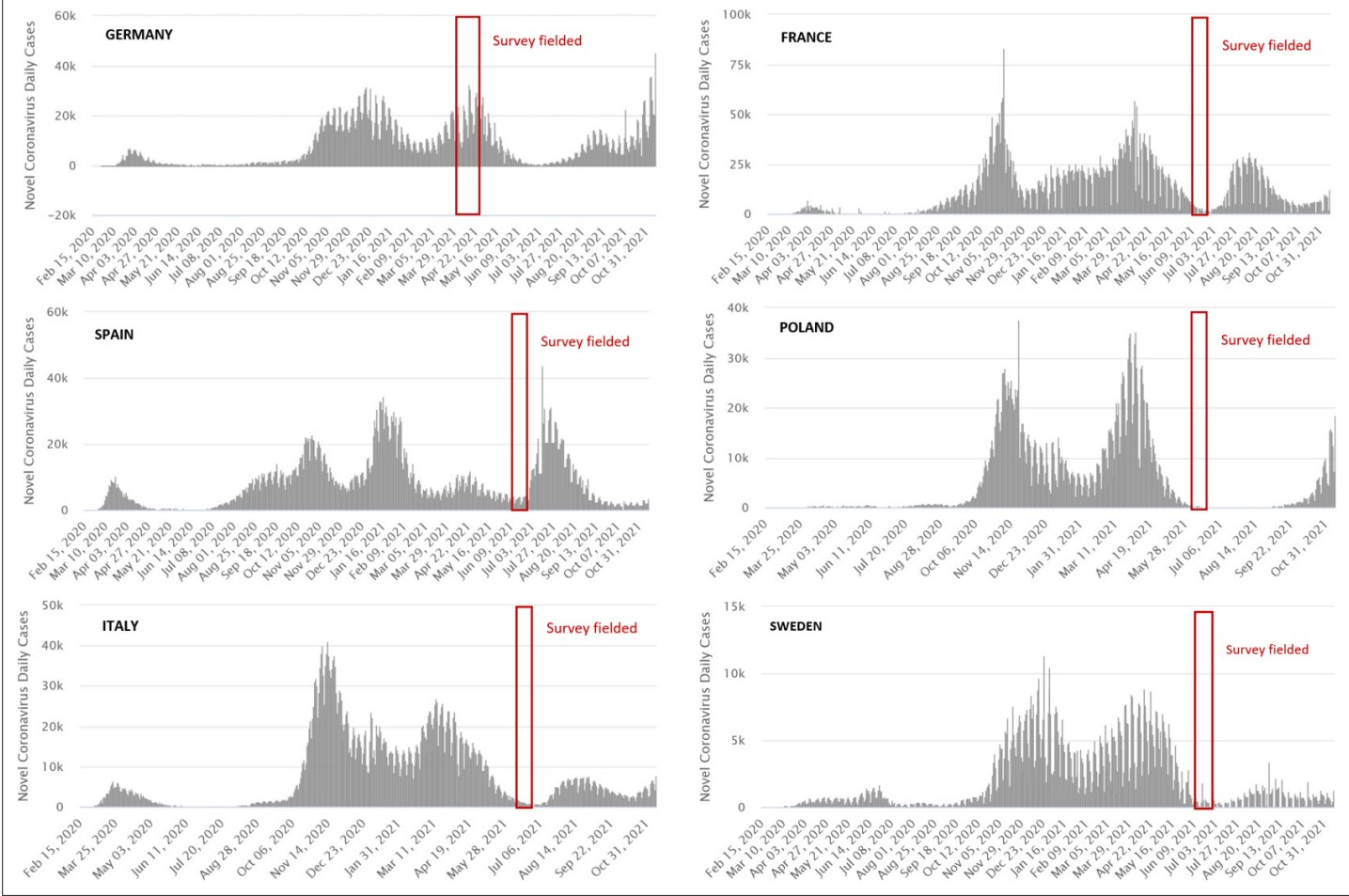

**Figure 1.** Timeline survey launch and infection rates. Source for COVID-19 daily case data: *Worldometers.info, 2021*.

The online version of this article includes the following figure supplement(s) for figure 1:

**Figure supplement 1.** COVID-19 threat perception across countries.

## Survey experiment

Discrete choice experiments (DCEs) are used to measure the relative importance of different characteristics that respondents weigh against each other when making certain choices (*Mangham et al., 2009*). DCEs have emerged from the theoretical tradition of random utility theory and are based on the assumption that respondents express their preferences by choosing the alternative associated with the highest individual benefits or utility (*Hall et al., 2004*). DCEs have advantages over other stated preference techniques, such as ranking or rating exercises, because they (i) more closely mimic real-world choice scenarios, (ii) reduce the cognitive complexity for respondents, and (iii) can elicit implicit preferences (*Mangham et al., 2009*).

We used the DCE methodology to elicit participants' preferences on the allocation of a COVID-19 vaccine dose to a person in the Global North or to a person in the Global South. We presented eight different choice sets and asked respondents to choose whether Person A or Person B should receive the COVID-19 vaccine first. Respondents were told that the other candidate in each pair would have to wait substantially longer to receive their first vaccine dose. In each of the eight choice sets, one candidate was described as living in the country of residence of the respondent – i.e. a high-income country with high healthcare system capacity, - and the other candidate was described as living in a low-income country with a low healthcare system capacity. The healthcare system capacity was explicitly mentioned in the two candidate profiles. Across choice sets, candidates varied along three additional attributes, namely (1) age (20; 40; 60; 80 years), (2) individual COVID-19 mortality risk due to comorbidity and/or lifestyle (no increased risk; increased risk; strongly increased risk), and

**Which of the following two persons should receive the vaccine first, person A or person B?**

| | Person A | Person B |
|---|---|---|
| **Age** | 20 years old | 40 years old |
| **Risk of COVID-19 death** | Increased risk due to comorbidity and/or lifestyle | No increased risk due to comorbidity and/or lifestyle |
| **Employment status** | Employed in essential services (e.g. health personel, supermarket employee) | Employed, and income losses due to COVID-19 restrictions |
| **Country of residence and health care system capacity** | Low-income country, with low healthcare system capacity (e.g. India, Nigeria, Bolivia) | Germany, with high healthcare system capacity |

| | Person A | Person B |
|---|---|---|
| **Your decision:** | ○ | ○ |

**Figure 2.** Sample choice set presented to respondents. Technical notes: The design was determined to be D-efficient based on weak priors for the main attributes effects (without interactions). Statistical efficiency was measured by the D-optimality criterion (D-error), the most widely used metric in this regard. D-optimal or D-efficient designs minimise the determinant of the asymptotic variance-covariance matrix, ensuring minimum variation around the parameter estimates.

(3) employment status (not employed; employed with a guaranteed income; employed and income losses due to COVID-19 restrictions; employed in essential services).

The specific combination of candidate profiles in the eight choice sets, that is the experimental design, was selected for statistical efficiency (referred to as 'D-efficiency'), which is accomplished by minimising the asymptotic variance-covariance matrix of the parameter estimates (based on an algorithm implemented in the software Ngene) (*Reed Johnson et al., 2013*). A sample choice situation as presented to respondents in the online survey, along with further technical information, is shown in *Figure 2* and notes.

Participants were only eligible to participate in the survey experiment if they (1) had not yet been vaccinated against COVID-19 at the time of the survey and (2) indicated that they were willing to get vaccinated. We employed these two criteria to elicit prioritisation preferences on how to distribute scarce COVID-19 vaccine doses among individuals who perceived the vaccine as beneficial *and* were themselves still waiting to receive their first dose.

Power calculations for the discrete choice experiment indicated an optimal sample size of n=2,061 respondents per country (*de Bekker-Grob et al., 2015*). However, the rapidly evolving pandemic situation in combination with conditional eligibility on vaccination status (see above) only allowed us to reach the required sample size in Germany due to the earlier timing of the data collection as well as budget constraints. Sample sizes in the remaining countries ranged from 670 in Poland to 925 in Spain, resulting in a total sample size of 6030 eligible responses.

## Heterogeneity variables

Similar to previous studies (*Duch et al., 2021*; *Persad et al., 2021*), we examined the effect of the candidate attribute 'Country of residence' for heterogeneity in terms of respondents' socioeconomic characteristics. Specifically, we assessed whether respondents' (i) gender, (ii) age, (iii) education, (iv) COVID-19 risk status (i.e. due to medical history), (v) COVID-19 threat perception, and (vi) employment status predicted differences in their allocation choices. Considering the limited statistical power for within-country analyses, the heterogeneity analysis was primarily conducted with the pooled sample. However, in order to gain a more thorough understanding of observed patterns, we also

report results for each country separately (bearing in mind the lack of statistical power in their interpretation). Moreover, we conducted an additional heterogeneity analysis to explore the relevance of the varying COVID-19 infection rates across countries at the time of the data collection. Specifically, we utilised subnational data from the European Centre for Disease Prevention and Control about the number of notified COVID-19 cases and assessed whether it predicts differences in respondents' vaccine allocation choices in the DCE.

## Statistical analyses

The empirical analysis comprised two steps. First, we estimated the main effects model for each country separately to assess the impact of the four candidate attributes (country of residence, age, COVID-19 mortality risk, employment status) on the probability of choosing a specific candidate. The statistical model we utilised was a conditional logit model (with respondent fixed-effects) with standard errors clustered at the respondent level. Thus, in the main effects model, we regressed respondents' vaccine allocation choice on the attribute levels of the respective candidate in each scenario. The potential impact of respondents' *own* characteristics was controlled for and absorbed by the fixed effects, given that respondents' characteristics do not vary across the eight choice sets of the DCE.

Second, we examined heterogeneity in the effect of the candidates-country-of-residence attribute by adding interaction terms between country of residence and heterogeneity variables to the above regression. Thus, the interaction of the respective heterogeneity variable (e.g. sex or age category) with the country of residence attribute - which varies across choice sets - allows us to examine the importance of respondents' characteristics, while still using a fixed-effects model. A pre-analysis plan along with justifications of any deviations thereof is accessible via https://osf.io/72jrq/.

## Ethical approval

The study received approvals from the ethics committees of the medical faculty at the Technical University of Munich (TUM, IRB 227/20 S) and the ethics board at the University of Trento (Trento, IRB 2021–027). Participants were given an individual link to the survey, where they first received information about the study's purpose, data protection regulations, and voluntary participation. All participants provided written electronic consent to participate in the study prior to commencing the survey. Personally identifying information such as names and contact details were not collected and data is thus fully anonymised. After completing the survey, participants received a voucher worth three to five Euros, which was distributed by the survey company.

# Results

## Sample characteristics

A total of 6030 eligible participants across all six countries completed the DCE. *Table 1* presents socioeconomic characteristics of the participants in each country. The German sample shows higher proportions of (i) older participants (age groups 55–64 and 65+years), (ii) less educated participants, and (iii) participants with an increased risk of a severe COVID-19 infection. Cross-country differences in the sample compositions and deviations from the census statistics are probably linked to the timing of the survey launch and the vaccination progress in each country. The German survey was launched earlier, when a higher number of older and high-risk inhabitants were not yet eligible for the COVID-19 vaccination and therefore were eligible to participate in the survey.

## Discrete choice experiment

*Table 2* lists the eight choice sets and presents the descriptive results of the discrete choice experiment, that is for each candidate, it presents the proportion of respondents who (in the given pair) chose that candidate over the other to receive the vaccine first.

*Table 3* summarises the results for the main effects model, which shows, separately for each country, the impact of the four attributes and levels on a candidate's likelihood of being chosen by the respondent to receive the COVID-19 vaccine first. In France, Italy, Spain, and Sweden, respondents, on average, chose the hypothetical candidate from the Global South over the hypothetical candidate from their own country to receive the vaccine first (Spain: OR: 1.79, 95% CI: 1.55–2.06; Italy: OR: 1.74, 95% CI: 1.50–2.01; Sweden: OR: 1.43, 95% CI: 1.24–1.65; France: OR: 1.37, 95% CI: 1.18–1.59;

**Table 1.** Socio-demographic characteristics.

| | Germany | Spain | Italy | France | Poland | Sweden |
|---|---|---|---|---|---|---|
| Female | 938 | 471 | 363 | 376 | 365 | 465 |
| | (47.76%) | (50.92%) | (47.33%) | (49.09%) | (54.48%) | (49.57%) |
| **Age group** | | | | | | |
| 18–24 | 148 | 111 | 107 | 106 | 185 | 104 |
| | (7.54%) | (12.00%) | (13.95%) | (13.84%) | (27.61%) | (11.09%) |
| 25–34 | 279 | 259 | 163 | 191 | 179 | 268 |
| | (14.21%) | (28.00%) | (21.25%) | (24.93%) | (26.72%) | (28.57%) |
| 35–44 | 361 | 307 | 162 | 217 | 152 | 281 |
| | (18.38%) | (33.19) | (21.12%) | (28.33%) | (22.69%) | (29.96%) |
| 45–54 | 376 | 193 | 177 | 236 | 105 | 179 |
| | (19.14%) | (20.86%) | (23.08%) | (30.81%) | (15.67%) | (19.08%) |
| 55–64 | 484 | 52 | 130 | 16 | 41 | 97 |
| | (24.64%) | (5.62%) | (16.95%) | (2.09%) | (6.12%) | (10.34%) |
| 65+ | 316 | 3 | 4 | 0 | 8 | 9 |
| | (16.09%) | (0.32%) | (0.52%) | (0%) | (1.19%) | (0.96%) |
| **Education** | | | | | | |
| Primary | 631 | 3 | 27 | 9 | 17 | 40 |
| | (32.13%) | (0.32%) | (3.52%) | (1.17%) | (2.54%) | (4.26%) |
| Secondary | 622 | 171 | 141 | 79 | 130 | 311 |
| | (31.67%) | (18.49%) | (18.38%) | (10.31%) | (19.40%) | (33.16%) |
| Higher | 359 | 264 | 353 | 301 | 337 | 159 |
| | (18.28%) | (28.54%) | (46.02%) | (39.30%) | (50.30%) | (16.95%) |
| University degree | 352 | 487 | 222 | 377 | 186 | 428 |
| | (17.92%) | (52.56%) | (28.94%) | (49.22%) | (27.76%) | (45.63%) |
| Employed | 1,204 | 647 | 480 | 579 | 460 | 753 |
| | (61.30%) | (69.95%) | (62.58%) | (75.59%) | (68.66%) | (80.28%) |
| CV-19 high-risk group | 1,041 | 255 | 204 | 219 | 256 | 190 |
| | (53.00%) | (27.57%) | (26.60%) | (28.59%) | (38.21%) | (20.26%) |
| Elevated CV-19 threat perception | 1,056 | 491 | 434 | 349 | 222 | 462 |
| | (53.88%) | (53.08%) | (58.41%) | (45.56%) | (33.13%) | (49.25%) |
| Observations | 1964 | 925 | 767 | 766 | 670 | 938 |

The table shows sample characteristics of participants by country, reporting both the absolute number of participants as well as the relative proportion of the respective characteristic prevalent in the sample. 'CV-19' denotes 'COVID-19'.

all p-values <0.001). For German respondents, we observe the opposite pattern: candidates from the Global South had significantly lower odds of being chosen to receive the vaccine (OR: 0.69, 95% CI: 0.62–0.76, p-value <0.001). In Poland, a candidate's country of residence neither increased nor decreased the odds of being chosen to receive the vaccine (OR: 0.99, 95% CI: 0.86–1.15).

For the attribute of COVID-19 mortality risk, we observe a similar pattern in all surveyed countries: The odds of being chosen to receive the vaccine were between two and almost six times higher for

**Table 2.** Full list of choice sets of the DCE and observed respondents' choices.

| Choice set | Age in years | COVID-19 mortality risk | Employment status | Country of residence and healthcare system capacity | Share of respondents who selected Person A and Person B within the given choice set (see col. 1) | | | | | | |
|---|---|---|---|---|---|---|---|---|---|---|---|
| | | | | | Pooled | Germany | Spain | Italy | France | Poland | Sweden |
| 1 Person A | 40 | No increased risk due to comorbidity and/or lifestyle | Not employed | *[Respondents' country of residence], with high healthcare system capacity | 16.77% | 16.70% | 14.70% | 19.04% | 15.14% | 24.93% | 12.58% |
| 1 Person B | 40 | Strongly increased risk due to comorbidity and/or lifestyle | Employed and guaranteed income | Low-income country, with poor healthcare system capacity | 83.23% | 83.23% | 85.30% | 80.96% | 84.86% | 75.07% | 87.42% |
| 2 Person A | 60 | Strongly increased risk due to comorbidity and/or lifestyle | Employed and guaranteed income | *[Respondents' country of residence], with high healthcare system capacity | 57.41% | 63.44% | 46.70% | 51.76% | 58.62% | 58.96% | 57.89% |
| 2 Person B | 60 | No increased risk due to comorbidity and/or lifestyle | Employed in essential services | Low-income country, with poor healthcare system capacity | 42.59% | 36.56% | 53.30% | 48.24% | 41.38% | 41.04% | 42.11% |
| 3 Person A | 60 | Increased risk due to comorbidity and/or lifestyle | Employed and income losses due to COVID-19 restrictions | *[Respondents' country of residence], with high healthcare system capacity | 57.06% | 68.74% | 45.51% | 46.94% | 58.49% | 55.22% | 52.45% |
| 3 Person B | 80 | Increased risk due to comorbidity and/or lifestyle | Not employed | Low-income country, with poor healthcare system capacity | 42.94% | 31.26% | 54.49% | 53.06% | 41.51% | 44.78% | 47.55% |
| 4 Person A | 20 | Increased risk due to comorbidity and/or lifestyle | Employed in essential services | Low-income country, with poor healthcare system capacity | 73.53% | 74.59% | 75.78% | 67.80% | 67.80% | 61.19% | 81.66% |
| 4 Person B | 40 | No increased risk due to comorbidity and/or lifestyle | Employed and income losses due to COVID-19 restrictions | *[Respondents' country of residence], with high healthcare system capacity | 26.47% | 25.41% | 24.22% | 32.20% | 32.20% | 38.81% | 18.34% |
| 5 Person A | 40 | No increased risk due to comorbidity and/or lifestyle | Employed in essential services | *[Respondents' country of residence], with high healthcare system capacity | 52.16% | 63.85% | 51.24% | 46.94% | 74.67% | 47.31% | 44.67% |
| 5 Person B | 20 | Increased risk due to comorbidity and/or lifestyle | Not employed | Low-income country, with poor healthcare system capacity | 47.84% | 36.15% | 48.76% | 53.06% | 25.33% | 52.69% | 55.33% |
| 6 Person A | 20 | Strongly increased risk due to comorbidity and/or lifestyle | Employed and income losses due to COVID-19 restrictions | Low-income country, with poor healthcare system capacity | 79.98% | 79.89% | 85.51% | 77.18% | 80.55% | 68.51% | 84.75% |
| 6 Person B | 20 | No increased risk due to comorbidity and/or lifestyle | Employed and guaranteed income | *[Respondents' country of residence], with high healthcare system capacity | 20.02% | 20.11% | 14.49% | 22.82% | 19.45% | 31.49% | 15.25% |
| 7 Person A | 80 | Increased risk due to comorbidity and/or lifestyle | Not employed | Low-income country, with poor healthcare system capacity | 42.70% | 31.82% | 52.00% | 53.98% | 41.64% | 49.40% | 43.18% |
| 7 Person B | 60 | Increased risk due to comorbidity and/or lifestyle | Employed and income losses due to COVID-19 restrictions | *[Respondents' country of residence], with high healthcare system capacity | 57.30% | 68.18% | 48.00% | 46.02% | 58.36% | 50.60% | 56.82% |
| 8 Person A | 40 | No increased risk due to comorbidity and/or lifestyle | Employed and guaranteed income | Low-income country, with poor healthcare system capacity | 14.49% | 7.89% | 13.19% | 29.07% | 15.93% | 22.54% | 10.77% |
| 8 Person B | 40 | Strongly increased risk due to comorbidity and/or lifestyle | Employed in essential services | *[Respondents' country of residence], with high healthcare system capacity | 85.51% | 92.11% | 86.81% | 70.93% | 84.07% | 77.46% | 89.23% |

*[Respondents' country of residence] was differed depending on the respective country in which the survey was fielded. E.g. in the German sample, this attribute level was 'Germany, with high healthcare system capacity'. The design was determined with a built-in constraint for the attributes Age and Employment status in order to avoid implausible combinations (specifically, an age of 80 was always combined with not being employed).

a candidate with an increased COVID-19 mortality risk, relative to a candidate with an average risk. The effect was even more pronounced for a candidate with a strongly increased mortality risk (ranging from OR: 4.35, 95% CI: 3.65–5.19 in Poland to OR: 14.20, 95% CI: 11.75–17.17 in Sweden; all p-values <0.001).

For employment status and age, we also observed largely similar patterns across countries: First, employed candidates who lost income due to the pandemic and candidates who were employed in essential services were significantly more likely to be chosen to receive the vaccine when compared to unemployed candidates. Second, in all countries, 40-year-old candidates had slightly higher odds of being chosen to receive the vaccine than 20- or 60-year-old candidates.

**Table 3.** Main attribute effects by country.

| | Germany | Spain | Italy | France | Poland | Sweden |
|---|---|---|---|---|---|---|
| | (1) | (2) | (3) | (4) | (5) | (6) |
| **Age** | | | | | | |
| 20 years (Reference category) | | | | | | |
| 40 years | 1.39*** | 1.25*** | 1.16*** | 1.15*** | 1.08* | 1.13*** |
| | [1.33,1.45] | [1.17,1.33] | [1.09,1.22] | [1.08,1.22] | [1.02,1.14] | [1.06,1.20] |
| 60 years | 0.87*** | 0.85*** | 0.88*** | 0.87*** | 0.85*** | 0.77*** |
| | [0.83,0.90] | [0.80,0.91] | [0.84,0.94] | [0.82,0.93] | [0.80,0.91] | [0.73,0.82] |
| 80 years | 0.81*** | 1.23** | 1.12 | 0.59*** | 0.94 | 0.70*** |
| | [0.73,0.90] | [1.06,1.43] | [0.94,1.33] | [0.50,0.69] | [0.78,1.13] | [0.61,0.81] |
| **COVID-19 mortality risk** | | | | | | |
| Average (Reference category) | | | | | | |
| Increased | 5.63*** | 3.05*** | 2.27*** | 4.54*** | 2.40*** | 5.67*** |
| | [5.13,6.19] | [2.66,3.50] | [1.96,2.64] | [3.88,5.31] | [2.07,2.79] | [4.91,6.56] |
| Strongly increased | 13.23*** | 9.68*** | 4.36*** | 9.04*** | 4.35*** | 14.20*** |
| | [11.64,15.03] | [8.12,11.53] | [3.69,5.15] | [7.50,10.91] | [3.65,5.19] | [11.75,17.17] |
| **Employment situation** | | | | | | |
| Not employed (Reference category) | | | | | | |
| Employed (guaranteed income) | 1.30*** | 0.79*** | 1.14 | 1.04 | 1.06 | 1.00 |
| | [1.18,1.42] | [0.68,0.92] | [0.99,1.32] | [0.89,1.22] | [0.91,1.24] | [0.87,1.16] |
| Employed (income losses) | 2.37*** | 1.91*** | 1.67*** | 1.65*** | 1.36*** | 1.73*** |
| | [2.19,2.56] | [1.71,2.13] | [1.49,1.87] | [1.47,1.86] | [1.21,1.53] | [1.54,1.94] |
| Essential services | 6.01*** | 4.05*** | 2.27*** | 2.74*** | 1.88*** | 4.22*** |
| | [5.38,6.72] | [3.50,4.70] | [1.98,2.59] | [2.33,3.22] | [1.65,2.15] | [3.57,4.99] |
| **Country of residence** | | | | | | |
| Respondents' country (Reference category) | | | | | | |
| Global South | 0.69*** | 1.79*** | 1.74*** | 1.37*** | 0.99 | 1.43*** |
| | [0.62,0.76] | [1.55,2.06] | [1.50,2.01] | [1.18,1.59] | [0.86,1.15] | [1.24,1.65] |
| Log likelihood | −14428.91 | −7090.45 | −6541.07 | −5994.74 | −5860.07 | −6875.04 |
| AIC | 28875.82 | 14198.89 | 13100.14 | 12007.48 | 11738.15 | 13768.07 |
| BIC | 28951.02 | 14267.31 | 13166.88 | 12074.21 | 11803.67 | 13836.62 |
| Pseudo $R^2$ | 0.22 | 0.19 | 0.10 | 0.17 | 0.08 | 0.23 |
| Observations | 31424 | 14800 | 12272 | 12256 | 10720 | 15008 |

Coefficients are odds ratios based on conditional logit estimations (respondent-level fixed effects) with standard errors clustered at the respondent level. Results to be interpreted relative to the indicated reference category, that is in the case of 'Country of residence', relative to the preference for the vaccine being given to a person living in the country of the survey respondent answering the question. 95% confidence intervals in brackets. *p<0.05, **p<0.01, ***p<0.001. Sample sizes reflect the eight choice tasks performed by each respondent.

Moving beyond the main attribute effects, *Figure 3* illustrates the results of the pooled heterogeneity analysis regarding respondents' socioeconomic characteristics (see *Supplementary file 1b, c* for exact coefficients by subgroup and interaction terms). The odds of choosing the candidate from the Global South rather than the candidate from the respondents' own country (all else equal) were significantly higher for female (OR of interaction: 1.22, 95% CI: 1.10–1.36, p-value <0.001)

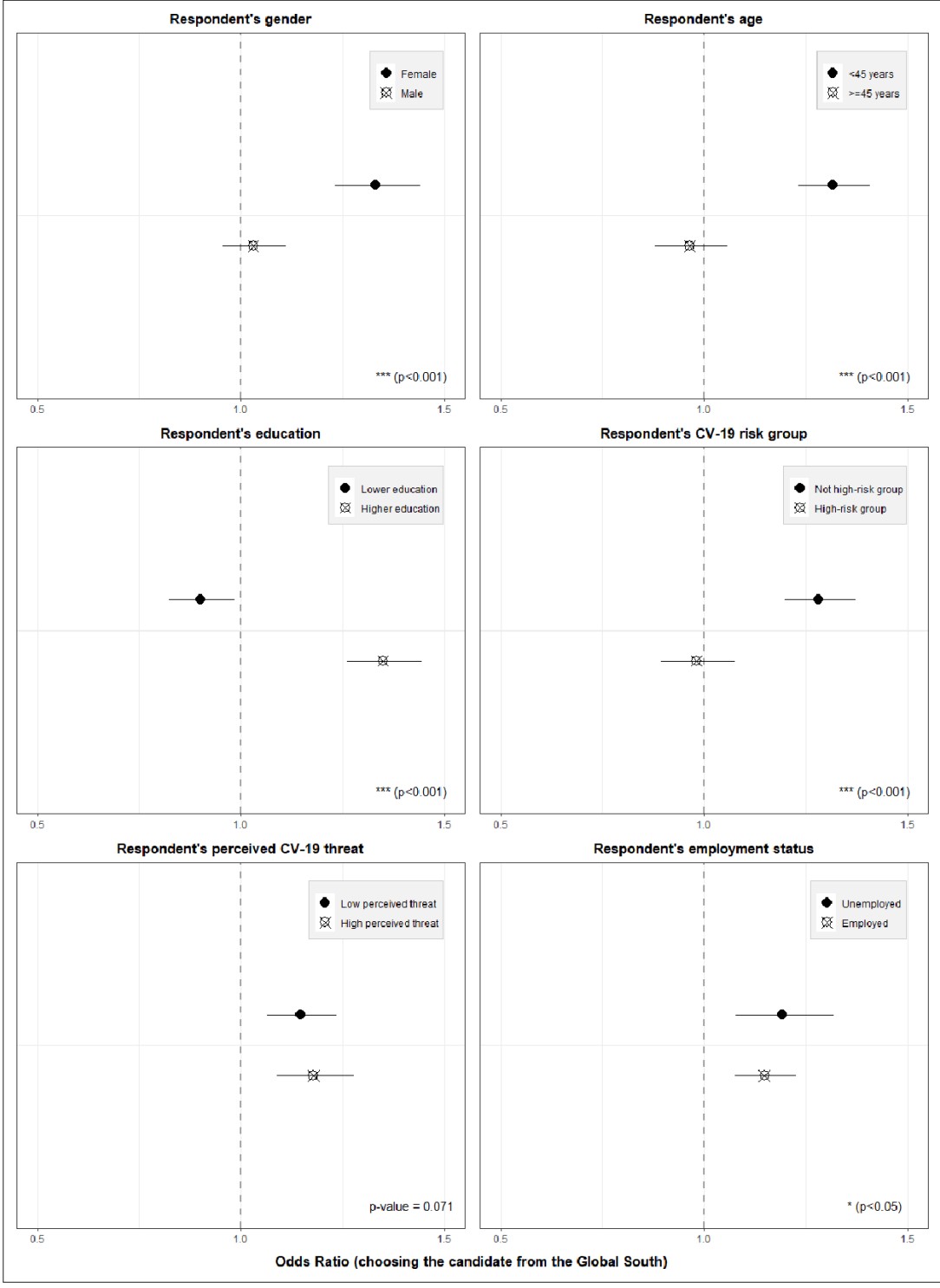

**Figure 3.** Heterogeneity of country attribute by respondent characteristics (pooled sample). Notes: Outcome: Choosing the respective candidate to receive the vaccine. Coefficients are odds ratios based on conditional logit estimations (respondent-level fixed effects) with standard errors clustered at the respondent level. Estimations were conducted controlling for the main effects of the other three attributes, but only the results for the candidate's country of residence attribute are shown here. Results to be interpreted relative to the indicated reference category, i.e. relative to the vaccine being given to a person living in the country of the survey respondent. The p-values on the bottom right of each graph indicates the statistical (in-)significance of the interaction term of the heterogeneity variable in question. For exact coefficients and interaction terms, see *Supplementary file 1b, c*.

and more educated (OR of interaction: 1.63, 95% CI: 1.46–1.82, p-value <0.001) respondents, and significantly lower for older respondents (above 45 years) (OR of interaction: 0.68, 95% CI: 0.61–0.76, p-value <0.001). Further, respondents who were themselves at high risk of a severe COVID-19 disease progression were significantly less supportive of distributing the COVID-19 vaccine to a candidate from the Global South (OR of interaction: 0.70, 95% CI: 0.63–0.78, p-value <0.001). In the subgroup of employed respondents, the odds of distributing the vaccine to a candidate in the Global South were slightly lower than in the subgroup of respondents that are not employed (OR of interaction: 0.89, 95% CI: 0.80–1.00, p-value = 0.045). The degree of COVID-19 threat perception did not seem to significantly affect respondents' distribution preferences (OR of interaction with a dummy variable indicating a high threat perception: 1.10, 95% CI: 0.99–1.22, p-value = 0.071).

*Figure 4* illustrates the extent to which these patterns prevail in each country (see *Supplementary file 1d–i* for exact coefficients of interaction terms). The heterogeneities we identified above were overall strongest in the German, Spanish, and Swedish samples. In the French and Italian sample, these patterns were only partly prevalent and largely not statistically significant. Interestingly, in terms of age, we found that older participants were relatively more supportive of distributing the vaccine to a candidate from the Global South in both France and Italy. In the Italian sample, this effect was even statistically significant (OR of interaction: 1.34, 95% CI: 1.00–1.78, p-value = 0.049). In the Polish survey, heterogeneity patterns were less clear, and we observed an opposite effect for educational attainment, with less educated respondents showing more pronounced preferences for an equitable vaccine distribution (OR of interaction: 0.72, 95% CI: 0.52–0.99, p-value = 0.044).

*Figure 5* and *Supplementary file 1j-k* present the results of the additional, pooled heterogeneity analysis using subnational data from the European Centre for Disease Prevention and Control on the number of notified COVID-19 cases during the time of data collection. When including the subnational COVID-19 incidence as a continuous heterogeneity variable, we found that respondents living in a region with a higher case incidence were on average slightly less supportive of distributing the vaccine to a candidate from the Global South (OR of interaction: 0.998, 95% CI: 0.997–0.998, p-value <0.001; see *Supplementary file 1k*). This effect is small in magnitude, but highly statistically significant. Based on an alternative specification, using a five-interval categorical variable for the incidence rate, we found that this heterogeneity seemed to be driven by a threshold incidence rate of more than 200–300 notified cases per 100.000 people, whereas it was less pronounced in the lower incidence regions and even showed the opposite pattern (i.e. more support with rising COVID-19 cases) when the case incidence is below 100. This finding remained stable both when using self-constructed intervals for the categorical variable (as shown in *Figure 3*) and when using quintiles informed by the distribution of the case incidence data itself (see *Supplementary file 1j-k*).

## Discussion

In our DCE conducted online in six EU countries, we found widespread global solidarity and support for a more equitable distribution of COVID-19 vaccines between the Global North and South. Our results suggest that vaccine allocation preferences are largely driven by the assessed vulnerability of possible recipients, irrespective of whether the recipient lives in the respondents' own country or in the Global South. Our respondents – who are themselves not yet vaccinated against COVID-19 but do favour vaccination – appear to rely upon the same solidarity considerations when deciding how vaccines should be distributed *within* a country and *between* countries. For example, they prioritise the candidate with the higher mortality risk but a foreign nationality over the candidate with a lower mortality risk but with the same nationality.

Interestingly, we found that respondents apply similar vulnerability considerations with regards to a candidate's employment status. Specifically, employed candidates who endured income losses due to the pandemic were significantly more likely to be chosen to receive the vaccine compared to candidates not employed. Thus, respondents seem to prioritise vaccine receipt according to the extent at which a candidate's employment status – and income generation – is affected by the pandemic situation. This is a finding, which, to the best of our knowledge, has not yet been identified explicitly by any of the other studies examining vaccine distribution preferences.

Importantly, we find that in a situation of vaccine scarcity and all else being equal, respondents from Spain, Italy, France and Sweden would prefer to allocate the vaccine to a person living in a country in the Global South with a worse health care system, as opposed to a person living in their

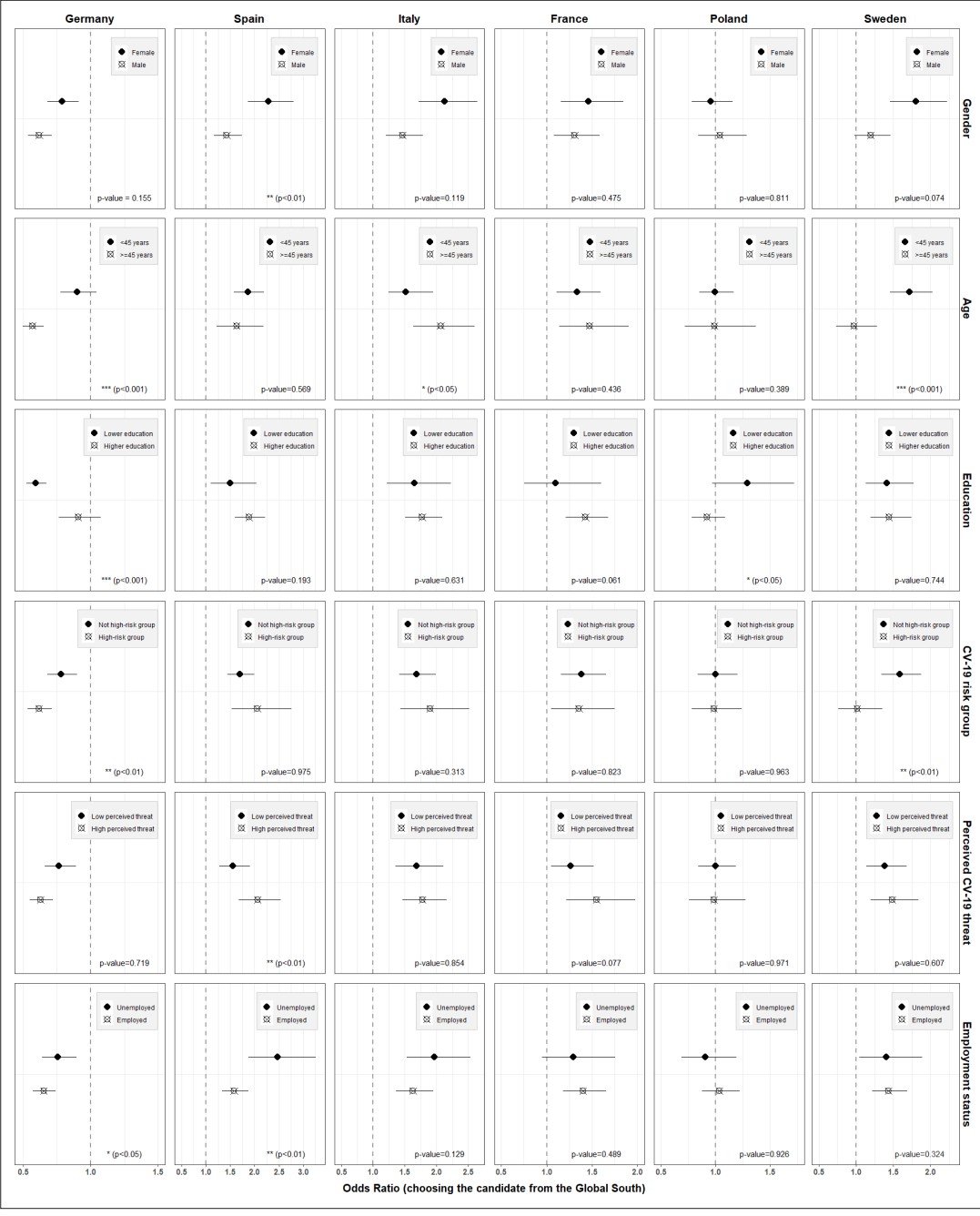

**Figure 4.** Heterogeneity of country attribute by respondent characteristics (country samples). Notes: Outcome: Choosing the respective candidate to receive the vaccine. Coefficients are odds ratios based on conditional logit estimations (respondent-level fixed effects) with standard errors clustered at the respondent level Estimations were conducted controlling for the main effects of the other three attributes, but only the results for the candidates' country of residence attribute are shown here. Results to be interpreted relative to the indicated reference category, i.e. in the case of country of residence, relative to the preference for the vaccine being given to a person living in the country of the survey respondent answering the question. The p-values on the bottom right of each graph indicates the statistical (in-)significance of the interaction term of the heterogeneity variable in question. For exact values of interaction terms, see **Supplementary file 1d–i**.

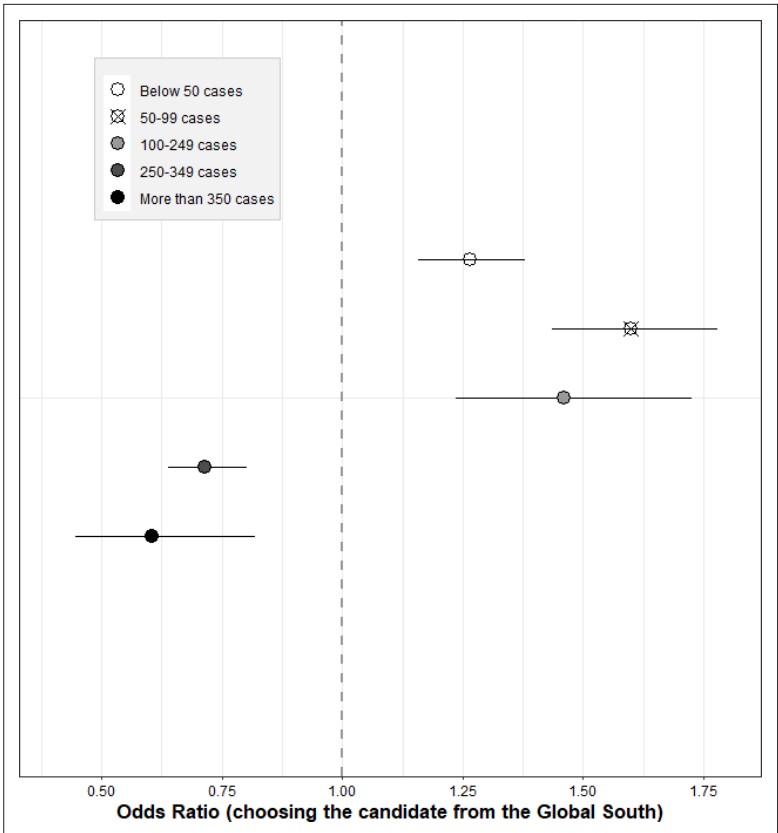

**Figure 5.** Heterogeneity of country attribute by regional COVID-19 case incidence (pooled sample). Notes: Outcome: Choosing the respective candidate to receive the vaccine. Coefficients are odds ratios based on conditional logit estimations (respondent-level fixed effects) with standard errors clustered at the respondent level. Data for the regional COVID-19 case incidence was drawn from https://www.ecdc.europa.eu/en/publications-data/weekly-subnational-14-day-notification-rate-covid-19 and reflects the 14-day notification rate of reported COVID-19 cases per 100 000 people, averaged over the data collection time period and sampled regions of the respective country. Estimations were conducted controlling for the main effects of the other three attributes, but only the results for the candidate's country of residence attribute are shown here. The coefficients are to be interpreted relative to the indicated reference category, i.e. relative to the vaccine being given to a person living in the country of the survey respondent. For exact coefficients, interaction terms, and p-values, see **Supplementary file 1j, k**.

own country of residence. We thus confirm the findings of the studies of citizens' distributional preferences that – *in the context of previous pandemics* – revealed largely positive attitudes towards vaccine donations to poorer countries (**Kumar et al., 2012**; **Ritvo et al., 2013**) and showed in experimental games that individuals (i) follow egalitarian motives in their own decisions (**Dawes et al., 2007**) and (ii) are even willing to punish third parties for inegalitarian behaviour (**Fehr and Gächter, 2002**). Yet, we go beyond previous studies by showing that such support for allocating vaccines to low-income countries with low healthcare system capacity holds across European countries from Spain to Sweden, even among respondents who had themselves not yet gotten vaccinated, and in the context of a pandemic with truly global reach, which by the time when we conducted our experiments had lasted for well more than a year.

At the same time, our DCE findings were inconclusive for the Polish and diverged for the German respondents: German survey participants were significantly less willing to allocate the COVID-19 vaccine to a person living in the Global South than to a person living in their own country, thus revealing preferences consistent with vaccine nationalism.

There are several possible explanations for the strikingly contrasting findings in Germany. First, studies have repeatedly found in-group bias (or here: national bias) in distributional preferences (**Bernhard et al., 2006**; **Reese et al., 2012**; **Yudkin et al., 2016**). Such in-group bias might explain

why German respondents prioritise, within their own country, candidates who are more vulnerable, for example due to facing higher COVID-19 risks, but not to candidates who are more vulnerable due to living in the Global South, if German participants had more nationalistic preferences, lower altruism, or less pro-social attitudes than participants from the other five European countries. We lack the data to test this possible explanation directly, but it seems unlikely: several cross-country analyses investigating nationalism, patriotism, and xenophobia have found German citizens to be among the least nationalistic in Europe (*Coenders et al., 2021*; *Lubbers and Coenders, 2017*). German citizens also do not generally appear to be outliers among their European peers in terms of altruism or similar pro-sociality variables (*Fehr et al., 2021*).

Second, respondents might have been influenced by heterogeneities in the pandemic situation across countries. In Germany, our experiment was implemented as the third pandemic wave peaked, whereas in the other countries, our DCEs were launched two months later, during a phase of relatively low daily case numbers. German respondents might therefore have felt a greater urgency about having faster access to vaccination. The evidence for this explanation is mixed, as our data does not show evidence of a substantially higher reported COVID-19 threat perception among German respondents (see *Supplementary file 1I* and *Figure 1—figure supplement 1*). Nevertheless, results from our subnational heterogeneity analysis provide some support for this explanation by suggesting that respondents residing in regions with a higher COVID-19 incidence rate during the time of data collection showed on average relatively less support for distributing the vaccine to a candidate from the Global South. Interestingly, this heterogeneity partly exhibits a quadratic functional form: respondents' support for distributing vaccine doses to the Global South slightly increases with COVID-19 case numbers in the lower parts of its distribution (<100 cases/100,000 people), but then more strongly decreases in the upper distribution, leading to an overall small, though highly statistically significant, negative interaction effect, which points to a decline in vaccine solidarity with low-income countries.

Third, the findings could be a function of the differences in the timing of the surveys relative to the progress of national vaccination campaigns. A recent survey experiment found that individuals with a higher perceived rank in the global income distribution feel stronger pressure to donate (*Fehr et al., 2021*). Extrapolating this dynamic to the context of our study, we might expect to see a higher individual inclination for vaccine donations in countries where the (perceived) vaccination rate is high by international standards. When the survey was fielded in Italy, France, Spain, Poland, and Sweden, the vaccination rate in those countries exceeded 50% (30%) for first (second) doses, whereas in Germany it was only at 25% (8%) (see *Supplementary file 1I*). This difference in domestic conditions may have affected the perceived scarcity of the COVID-19 vaccines between countries and the normative assessment thereof. In addition, the differential vaccination roll-out was likely also the reason for having been able to include a much higher proportion of older as well as high-medical-risk respondents in the German sample. Our separate heterogeneity analysis for Germany shows that especially the larger proportion of older respondents and high-risk respondents (the former more so than the latter), in combination with the German sample having a higher proportion of lower educated respondents, may have resulted in the lower global vaccine solidarity among German respondents (see *Figure 2* and *Supplementary file 1d*).

In view of the above, the differential findings for Germany are most likely driven by a combination of (i) higher infection and (ii) lower vaccination rates at the time of the German data collection and (iii) a higher proportion of older, high-risk and lower educated respondents in the German sample. While we are able to provide empirical support in favor of (i) and (iii), our analysis did not allow for any in-depth investigation of (ii), not least due to a lack of publicly available data on subnational vaccination rates for some of the countries.

A number of limitations are worth noting. First, in addition to the cross-country differences in COVID-19 infection and vaccination rates discussed above, participants' preferences may partly also be driven by cross-country variation in the pandemic trajectory over time. Given that our study utilises only cross-sectional survey data, we are not able to empirically examine any changes in public perceptions regarding the pandemic over time. Yet, the pronounced temporal nature of the pandemic needs to be kept in mind when reflecting on our findings. This includes, for instance, (i) notions of scarcity in the early phases of the vaccination campaigns, which may have affected perceptions about their value, (ii) a potential erosion of public trust during the suspension of the AstraZeneca/Johnson&Johnson vaccines, and (iii) increases in conspiracy beliefs related to vaccination through increasing

spread of misinformation via social media channels (*Steinert et al., 2022*). More generally, however, citizens' overall assessment of the pandemic situation likely varies strongly with the various stages of the pandemic, which we are at least partly able to capture by exploiting the different timing of our survey launches and thus some country (and regional) differences in infection rates at the time of data collection. Second, on an individual level, there may be additional characteristics that explain approval or rejection of COVID-19 vaccine donations but were not captured in all six countries, including (1) nationalistic attitudes, (2) altruistic preferences, or (3) migration background. Third, our analysis points to a number of predictors of variation. However, they are not susceptible to experimental manipulation and should therefore not be interpreted as causal. Fourth, at the country level (except in Germany), we were not able to achieve our initial target sample sizes due to budget constraints; sample sizes therefore also varied across countries. We surely have adequate statistical power in the pooled analysis, but with the exception of Germany might have insufficient statistical power in the country-specific heterogeneity analyses. Findings indicating, for some countries, a lack of statistical significance at conventional levels should therefore be interpreted with caution. That said, since we find statistically significant effects of the main attributes in all countries except from Poland, lack of statistical power might have been less of a concern in the main-effects analysis. Finally, as a consequence of the two eligibility criteria of (1) not yet having been vaccinated against COVID-19 and (2) being willing to get vaccinated, our samples are not nationally representative of the population profile of each country (compare *Table 1* with the census data in *Supplementary file 1a*). However, since we initially sampled participants based on quotas that were matched to the census population of each target country in terms of gender, age, education, and geographic location, our samples should be representative of each country's unvaccinated and willing-to-get-vaccinated population at the time of the data collection regarding these quotas. In view of this, our suggested estimates of citizens' support for global solidarity in distributing the COVID-19 vaccines are most likely lower bound estimates – assuming that already vaccinated citizens and those less interested in getting vaccinated would be even more willing to donate vaccine doses to the Global South.

Policymakers and global health scholars have condemned the unequitable distribution of COVID-19 vaccines between high- and low-income countries as 'vaccine apartheid' (*Gonsalves and Yamey, 2021*; *Harman et al., 2021*). Still, the World Health Organisation's call for a moratorium on COVID-19 booster vaccinations in high-income countries in favour of prioritising first dose vaccinations in low-income countries went unheeded, at least in part out of a sense that donating vaccines to countries in the Global South lacks popular support and might even subject the government to electoral punishment (*Krause et al., 2021*). The emergence of the Omicron and its various subvariants throughout 2022, as well as ongoing concerns about potential future variants, emphasise once again the transboundary nature of the COVID-19 pandemic. Acknowledging this characteristic of pandemics, governments in high-income countries should discard mitigation strategies that are guided by the premise of vaccine nationalism (*Vanhuysse et al., 2021*). Findings from our study suggest that governments of European countries can rely on solid public approval for a more equitable vaccine distribution – especially among female, younger and more educated citizens. Public support for vaccine donations to the Global South may be even more pronounced at the time of writing, since COVID-19 risk groups have now already received their first (or even second) booster vaccination. Moreover, our finding regarding prioritising candidates according to employment status, and particularly potential income losses, may also indirectly point to a pronounced public support for vaccine donations to the Global South: Employment security is much lower in low- and middle-income countries, especially in view of the high share of workers in the informal sector, where substantial income losses are more likely to occur during a lockdown or pandemic-induced economic downturns. More effective international policy initiatives to ensure efficient, adequate, and timely COVID-19 vaccine transfers to low-income countries are therefore urgently needed. They likely remain highly relevant and timely in view of the ongoing development of new, variant-specific vaccine updates such as those for the Omicron variant.

## Acknowledgements

We thank everyone who helped with translating: Walter Osika, Jocelyn Raude, Jonathan Garcia Fuentes, Anna Glyk, and Kathrin and Michal Bartoszewski. This project was funded by the European Union's Horizon 2020 research and innovation programme under grant agreement No 101016233 (PERISCOPE). We are grateful for helpful comments on the analysis and interpretation from Philipp

Lergetporer, Michael Kurschilgen and Abu Siddique, Nils Weidmann, Dirk Leuffen, and Gerald Schneider.

## Additional information

### Funding

| Funder | Grant reference number | Author |
| --- | --- | --- |
| Horizon 2020 Framework Programme | No 101016233 (PERISCOPE) | Janina I Steinert Giuseppe A Veltri Tim Büthe |

The funders had no role in study design, data collection and interpretation, or the decision to submit the work for publication.

### Author contributions

Janina I Steinert, Conceptualization, Data curation, Supervision, Funding acquisition, Validation, Investigation, Visualization, Methodology, Writing – original draft, Writing – review and editing; Henrike Sternberg, Conceptualization, Data curation, Formal analysis, Validation, Visualization, Methodology, Writing – original draft, Project administration, Writing – review and editing; Giuseppe A Veltri, Conceptualization, Funding acquisition, Investigation, Methodology, Writing – review and editing; Tim Büthe, Conceptualization, Supervision, Funding acquisition, Methodology, Writing – review and editing

### Author ORCIDs

Janina I Steinert https://orcid.org/0000-0001-7120-0075
Henrike Sternberg https://orcid.org/0000-0001-8539-6478
Giuseppe A Veltri https://orcid.org/0000-0002-9472-2236
Tim Büthe https://orcid.org/0000-0002-4724-5000

### Ethics

Human subjects: The study received approvals from the ethics committees of the medical faculty at the Technical University of Munich (TUM, IRB 227/20 S) and the ethics board at the University of Trento (Trento, IRB 2021-027). Participants were given an individual link to the survey, where they first received information about the study's purpose, data protection regulations, and voluntary participation. All participants provided written electronic consent to participate in the study prior to commencing the survey. Personally identifying information such as names and contact details were not collected and data is thus fully anonymised. After completing the survey, participants received a voucher worth three to five Euros, which was distributed by the survey company.

### Decision letter and Author response

Decision letter https://doi.org/10.7554/eLife.79819.sa1
Author response https://doi.org/10.7554/eLife.79819.sa2

## Additional files

### Supplementary files

• Supplementary file 1. The supplementary file contains supplementary tables to further contextualise the results. (a) Population Census Data for Key Sociodemographic Variables. (b) Country of residence attribute main effect - By subgroups of respondent characteristics (pooled sample). (c) Country of residence attribute: Heterogeneity by respondent's characteristics (pooled results). (d) Country of residence attribute: Heterogeneity by respondent's characteristics (German sample). (e) Country of residence attribute: Heterogeneity by respondent's characteristics (Spanish sample). (f) Country of residence attribute: Heterogeneity by respondent's characteristics (Italian sample). (g) Country of residence attribute: Heterogeneity by respondent's characteristics (French sample). (h) Country of residence attribute: Heterogeneity by respondent's characteristics (Polish sample). (i) Country of residence attribute: Heterogeneity by respondent's characteristics (Swedish

sample). (j) Country of residence attribute main effect - By subgroups of regional case incidence (pooled sample). (k) Country of residence attribute: Heterogeneity by regional case incidence (pooled results). (l) Country level differences in case incidence, vaccination rates, willingness and threat perception

• MDAR checklist

### Data availability

All data generated or analysed during this study are made publicly available via the Open Science Framework under the following link: https://osf.io/72jrq/.

The following dataset was generated:

| Author(s) | Year | Dataset title | Dataset URL | Database and Identifier |
|---|---|---|---|---|
| Steinert S, Veltri B | 2021 | Public Opinion on the Distribution of COVID-19 Vaccines between the Global North and South | https://osf.io/72jrq/ | Open Science Framework, 10.17605/OSF.IO/72JRQ |

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
