## [Editor Report]

Despite all efforts, the global distribution, developed versus developing economies, of Covid-19 vaccines and immunisation with them remains highly unequal. This rigorous and well-designed study addresses the question of what are the public preferences for vaccine allocation globally in 6 European countries. The results documented in this paper show that overall the public preference is for equitable distribution and provide important evidence and insights for policymakers.

---

## [Decision Letter]

**Decision letter after peer review:**

Thank you for submitting your article "How Should COVID-19 Vaccines be Distributed between the Global North and South? A Discrete Choice Experiment in Six European Countries" for consideration by *eLife*. Your article has been reviewed by 1 peer reviewer, and the evaluation has been overseen by a Reviewing Editor and a Senior Editor. The reviewers have opted to remain anonymous.

As is customary in *eLife*, the reviewers have discussed their critiques with one another and with the Reviewing and Senior Editors. The decision was reached by consensus. What follows below is the Reviewing Editor's edited compilation of the essential and ancillary points provided by reviewers in their critiques and in their interaction post-review. Please submit a revised version that addresses these concerns directly. Although we expect that you will address these comments in your response letter, we also need to see the corresponding revision clearly marked in the text of the manuscript. Some of the reviewers' comments may seem to be simple queries or challenges that do not prompt revisions to the text. Please keep in mind, however, that readers may have the same perspective as the reviewers. Therefore, it is essential that you amend or expand the text to clarify the narrative accordingly.

Essential revisions:

1) The sample from Germany is noticeably different from the rest of the countries (particularly in terms of having a higher ratio of those who are in the high-risk category). This might have impacted the results and needs to be reflected in the study discussion. Also, there is heterogeneity between studies in terms of the time of the fieldwork and each country's conditions in regards to the vaccination roll-out and the number of infections at that time. A study limitation that needs to be noted is that despite the large sample sizes and the inclusion criterion of not being vaccinated, the samples are not nationally representative of the population profile of each country, hence, the results are not representative of the views of the entire population.

2. The manuscript narrative needs to be updated to reflect the present conditions in terms of inoculation campaigns, their success rate, and their disparities across the world. While the sense of urgency at the beginning of the global inoculation campaign seems to be passed, the inequity between the global north and south has remained persistent.

3. There is space for more discussions on an interesting finding of the study that is prioritizing the vaccines according to employment status, and in particular income loss.

4. The temporal nature of the public views at various stages of the pandemic and vaccination campaigns should also be noted.

5. During the analysis of the main results (Table 2), were respondents' characteristics, including being in a high-risk group, controlled for? It will be helpful to clarify this in the paper. Yet even with the inclusion of such controls, it is likely that the characteristics of the sample from Germany and the timing of the fieldwork during which Germany had high infection rates while being behind the rest of the EU countries in vaccination rates play a role in the study findings from Germany. A stark difference between country samples e.g. is the high rate of those respondents who were in the high-risk group in Germany. This factor, even after being controlled for, very much likely has influenced respondents' attitudes towards sharing the vaccines with the rest of the world.

6. It will be interesting to see a chart using the pooled data depicting the ratio of responses to the 8 choice sets of the DCE (as shown in Table 4).

7) This paper provides solid evidence on public opinion from six European countries on key attributes according to which they believe COVID-19 vaccines should be prioritized. The paper has valuable policy implications.

*Reviewer #1 (Recommendations for the authors):*

1. Given that the events since the start of the pandemic have rapidly evolved, the introduction and Discussion sections require updates. While the sense of urgency at the beginning of the global inoculation campaign seems to be passed, the inequity between the global north and south has remained persistent. I believe the narrative of the entire manuscript needs to be changed and perhaps it can be tailored in the context of the possibility of future pandemics when another global campaign becomes essential.

2. I trust that during the analysis of the main results (Table 2), respondents' characteristics, including being in a high-risk group, were controlled for? It will be helpful to clarify this in the paper.

3. Yet even with the inclusion of such controls, I believe that the characteristics of the sample from Germany and the timing of the fieldwork during which Germany had high infection rates while being behind the rest of the EU countries in vaccination rates play a role in the study findings from Germany. A stark difference between countries' samples e.g. is the high rate of those respondents who were in the high-risk group in Germany. This factor, even after being controlled for, very much likely has influenced respondents' attitudes towards sharing the vaccines with the rest of the world.

4. A study limitation that needs to be noted is that despite the large sample sizes and the inclusion criterion of not being vaccinated, the samples are not nationally representative of the population profile of each country, hence, the results are not representative of the views of the entire population.

5. It will be interesting to see a chart using the pooled data depicting the ratio of responses to the 8 choice sets of the DCE (as shown in Table 4).

---

## [Author Response]

Essential revisions:1) The sample from Germany is noticeably different from the rest of the countries (particularly in terms of having a higher ratio of those who are in the high-risk category). This might have impacted the results and needs to be reflected in the study discussion. Also, there is heterogeneity between studies in terms of the time of the fieldwork and each country's conditions in regards to the vaccination roll-out and the number of infections at that time. A study limitation that needs to be noted is that despite the large sample sizes and the inclusion criterion of not being vaccinated, the samples are not nationally representative of the population profile of each country, hence, the results are not representative of the views of the entire population.

Thank you very much for the important suggestion to address the divergent findings for the German sample more fully. We agree that the notably different composition of the German sample has likely impacted the study findings, i.e. the much lower support among German respondents for distributing vaccines to the Global South. Combining the information regarding the differences in the sample compositions across countries (Table 1) with the findings from the heterogeneity analysis within the German sample (first column of Figure 4, now added to the main part of the paper) suggests that especially the higher proportions of older, high medical-risk and lower educated respondents might explain why we observe substantially lower global solidarity among German respondents. We moreover agree that the differences in infection and vaccination rates across countries, and especially between Germany and the rest of the sampled countries (see Supplementary File 1l and Appendix 1 – Figure 1 in the supplementary material), likely also contributed to the differential results for Germany. In addition, both points are probably related, as the much lower vaccination rate in Germany, at the time at which the survey was fielded, allowed for the inclusion of a larger proportion of older and high-risk respondents, given our eligibility criterium of not yet having been vaccinated against COVID-19.

To address the above points, we have (i) extended our empirical analysis by adding an additional heterogeneity analysis and (ii) notably expanded the Discussion section of the paper to incorporate the above points in more detail.

Regarding (i): The additional heterogeneity analysis uses subnational data from the European Centre for Disease Prevention and Control on COVID-19 infection rates during the time of data collection in each included country. The analysis reveals that respondents residing in regions with a higher infection rate have on average a lower preference for distributing the vaccine to a candidate from the Global South. This provides some support for the supposition that the higher infection rates in Germany compared to the other countries might have played a crucial role also for the observed country-level differences in global solidarity. We would have liked to also include an additional heterogeneity analysis for subnational vaccination rates at the respective points in time, but, unfortunately, such data is not available for all of the six countries in our sample (including, crucially, not for Germany). To properly incorporate these additions into the empirical analysis, we added Figure 5 to the main manuscript, Supplementary Files 1j-k to the Supplementary Material, and new paragraphs to the Material and Methods section and to the Results sections of the main manuscript:

Additions to: Materials and Methods (Subsection: Heterogeneity Variables; last paragraph)

“Moreover, we conducted an additional heterogeneity analysis to explore the relevance of the varying COVID-19 infection rates across countries at the time of the data collection. Specifically, we utilised subnational data from the European Centre for Disease Prevention and Control about the number of notified COVID-19 cases and assessed whether it predicts differences in respondents’ vaccine allocation choices in the DCE.”

Additions to: Results

“Figure 5 and Supplementary Files 1j-k present the results of the additional, pooled heterogeneity analysis using subnational data from the European Centre for Disease Prevention and Control on the number of notified COVID-19 cases during the time of data collection. When including the subnational COVID-19 incidence as a continuous heterogeneity variable, we found that respondents living in a region with a higher case incidence were on average slightly less supportive of distributing the vaccine to a candidate from the Global South (OR of interaction: 0.998, 95% CI: 0.997-0.998, pvalue<0.001; see Supplementary File 1l). This effect is small in magnitude, but highly statistically significant. Based on an alternative specification, using a five-interval categorical variable for the incidence rate, we found that this heterogeneity seemed to be driven by a threshold incidence rate of more than 200-300 notified cases per 100.000 people, whereas it was less pronounced in the lower incidence regions and even showed an opposite pattern (i.e. more support with rising COVID-19 cases) when the case incidence is below 100. This finding remained stable both when using self-constructed intervals for the categorical variable (as shown in Figure 5) and when using quintiles informed by distribution of the case incidence data itself (see Supplementary Files 1jk).”

Regarding (ii): We have notably expanded the Discussion section of the paper in order to provide a more thorough reflection on the differential findings for Germany, now including all the points mentioned above as well as the new findings from the additional heterogeneity analysis of the COVID-19 case incidence. In line with this, we have also made the within country heterogeneity analyses a bit more prominent in the manuscript by moving the corresponding figure from the supplementary materials to the main manuscript (previously Figure S3, now Figure 4). We hope that this will make it easier for the reader to see why and which differences in the sample composition are likely driving the lower support among German citizens for distributing vaccines to the Global South. Please find below the additions made in the Discussion section of the paper:

Additions to: Discussion

“Second, respondents might have been influenced by heterogeneities in the pandemic situation across countries. In Germany, our experiment was implemented as the third pandemic wave peaked, whereas in the other countries, our DCEs were launched two months later, during a phase of relatively low daily case numbers. German respondents might therefore have felt a greater urgency about having faster access to vaccination – though our data does not show evidence of a substantially higher reported COVID-19 threat perception among German respondents (see Supplementary File 1l and Figure 1 —figure supplement 1). Nevertheless, results from our subnational heterogeneity analysis provide some support for this explanation by suggesting that respondents residing in regions with a higher COVID-19 incidence rate during the time of data collection showed, on average, relatively less support for distributing the vaccine to a candidate from the Global South. Interestingly, this heterogeneity partly exhibits a quadratic functional form: respondents’ support for distributing vaccine doses to the Global South slightly increases with COVID-19 case numbers in the lower parts of its distribution (<100 cases/100.000 people), but then more strongly decreases in the upper distribution, leading to an overall small though statistically significant negative interaction effect, which points to a decline in vaccine solidarity with low-income countries.

Third, the findings could be a function of the differences in the timing of the surveys relative to the progress of national vaccination campaigns. A recent survey experiment found that individuals with a higher perceived rank in the global income distribution feel stronger pressure to donate. (Fehr et al., 2021) Extrapolating this dynamic to the context of our study, we might expect to see a higher individual inclination for vaccine donations in countries where the (perceived) vaccination rate is high by international standards. When the survey was fielded in Italy, France, Spain, Poland, and Sweden, the vaccination rate in those countries exceeded 50% (30%) for the first (second) doses, whereas in Germany it was only at 25% (8%) (see Supplementary File 1j). This difference in domestic conditions may have affected the perceived scarcity of the COVID-19 vaccines between countries and the normative assessment thereof. In addition, the differential vaccination roll-out was likely also the reason for having been able to include a much higher proportion of older as well as high-medical-risk respondents in the German sample. Our separate heterogeneity analysis for Germany shows that especially the larger proportion of older respondents and high-risk respondents (the former more so than the latter), in combination with the German sample having a higher proportion of lower educated respondents, may have resulted in the lower global vaccine solidarity among German respondents (see Figure 4 and Supplementary File 1d).

In view of the above, the differential findings for Germany are most likely driven by a combination of (i) higher infection and (ii) lower vaccination rates at the time of the German data collection and (iii) a higher proportion of older, high-risk, and lower educated respondents in the German sample. While we are able to provide empirical support in favor of (i) and (iii), our analysis did not allow for any in-depth investigation of (ii), not least due to a lack of publicly available data on subnational vaccination rates for some of the countries.”

Finally, we also agree with your final point in this comment, i.e. that our samples are likely not representative of the national population in the sampled countries. We agree that it should be explicitly mentioned in the limitations of the paper. Accordingly, we have included the following note in our Discussion section:

Additions to: Discussion

“Finally, as a consequence of the two eligibility criteria of (1) not yet having been vaccinated against COVID-19 and (2) being willing to get vaccinated, our samples are not nationally representative of the population profile of each country (compare Table 1 with the census data in Supplementary File 1a). However, since we initially sampled participants based on quotas that were matched to the census population of each target country in terms of gender, age, education, and geographic location, our samples should be representative of each country’s unvaccinated and willing-to-get-vaccinated population at the time of the data collection regarding these quotas. In view of this, our suggested estimates of citizens’ support for global solidarity in distributing the COVID19 vaccines are most likely lower bound estimates – assuming that already vaccinated citizens and those less interested in getting vaccinated would be even more willing to donate vaccine doses to the Global South.”

2. The manuscript narrative needs to be updated to reflect the present conditions in terms of inoculation campaigns, their success rate, and their disparities across the world. While the sense of urgency at the beginning of the global inoculation campaign seems to be passed, the inequity between the global north and south has remained persistent.

Thank you very much for this important comment and we agree that the manuscript needs to be updated regarding these points, given that it reflected the current pandemic situation and state of the inoculation campaign at the time the initial manuscript was written (i.e. end of 2021). We have now updated the manuscript’s narrative by revising the introduction as well as the conclusion and have put a specific focus on the persistence of the global inequality in inoculation campaigns as well as in the more recent booster campaigns and expected future campaigns for an Omicron-specific vaccine.

Additions to: Introduction

“In his opening speech to address the United Nations General Assembly in September 2021, Secretary General António Guterres expressed stark discontent with the highly unequal global distribution of COVID-19 vaccines: “A majority of the wealthier world is vaccinated. Over 90 percent of Africans are still waiting for their first dose. This is a moral indictment of the state of our world. It is an obscenity”. (UN Secretary General, 2021). At the time of writing, and a year after Guterres gave his speech, 80% of the citizens of low-income countries are still waiting for their first dose of a COVID-19 vaccine, whereas in high-income countries, almost 80% of citizens are vaccinated. While the initial sense of urgency at the onset of the pandemic may have passed, this blatant inequity in access to COVID-19 vaccines prevails, continuing also throughout the more recent booster campaigns: In the past twelve months, the total number of vaccine doses administered per 100 people amounts to almost 100 in high-income countries and to only 27 in low-income countries.(Our World in Data; Mathieu et al., 2021) It therefore also seems likely that we will observe this same pattern again with the new Omicron-specific vaccine.

[…]

The globally unequal distribution of COVID-19 vaccines, including the stockpiling of vaccine doses for their own citizens, is partly a consequence of widespread vaccine nationalism in high-income countries.(Harman et al., 2021; Herzog et al., 2021; Wagner et al., 2021) To ensure “fair and equitable access” to COVID-19 vaccines for all countries and achieve high vaccination rates everywhere, the World Health Organisation (WHO), the Vaccine Alliance (Gavi), and the Coalition for Epidemic Preparedness Innovations (CEPI) formed a multilateral initiative named “COVID-19 Vaccines Global Access”, COVAX.(Herzog et al., 2021) However, several governments have resorted to making bilateral purchasing agreements with vaccine manufacturers outside of COVAX, which has substantially weakened the initiatives’ collective purchasing power.(Kim, 2021; Wouters et al., 2021) Moreover, out of those vaccine doses that were initially announced as donations to COVAX by high-income countries, substantial proportions – 25% of announced EU doses and almost 50% of announced US doses – have in fact not yet been donated (WHO ACT-Accelerator Hub; Our World in Data; Mathieu et al., 2021). The WHO’s pledges for a more equitable COVID-19 vaccine distribution have thus still not been fulfilled.”

Additions to: Discussion

“The emergence of the Omicron and its various subvariants throughout 2022, as well as ongoing concerns about potential future variants, emphasise once again the transboundary nature of the COVID-19 pandemic. Acknowledging this characteristic of pandemic, governments in high-income countries should discard mitigation strategies that are guided by the premise of vaccine nationalism. (Vanhuysse et al., 2021) Findings from our study suggest that governments of European countries can rely on solid public approval for a more equitable vaccine distribution – especially among female, younger and more educated citizens. Public support for vaccine donations to the Global South may be even more pronounced at the time of writing, since COVID-19 risk groups have now already received their first (or even second) booster vaccination. Moreover, our finding regarding prioritising candidates according to employment status, and particularly potential income losses, may also indirectly point to a pronounced public support for vaccine donations to the Global South: Employment security is much lower in low- and middle-income countries, especially in view of the high share of workers in the informal sector, where substantial income losses are more likely to occur during a lockdown or pandemic-induced economic downturns. More effective international policy initiatives to ensure efficient, adequate, and timely COVID-19 vaccine transfers to low-income countries are therefore urgently needed. They likely remain highly relevant and timely in view of the ongoing development of new, variant-specific vaccine updates such as those for the Omicron variant.”

3. There is space for more discussions on an interesting finding of the study that is prioritizing the vaccines according to employment status, and in particular income loss.

Thank you very much for this very helpful suggestion. We agree that this is an interesting and important finding, and also a rather novel one, which has not yet been extensively assessed or discussed in previous studies. We have revised the Discussion section in two places to make this finding more prominent, discuss its relevance against the existing literature, and emphasise its importance with regards to the overall narrative of the study:

Additions to: Discussion (first part)

“Interestingly, we found that respondents apply similar vulnerability considerations with regards to a candidate’s employment status. Specifically, employed candidates who endured income losses due to the pandemic were significantly more likely to be chosen to receive the vaccine compared to candidates not employed. Thus, respondents seem to prioritise vaccine receipt according to the extent at which a candidate’s employment status – and income generation – is affected by the pandemic situation. This is a finding, which, to the best of our knowledge, has not yet been identified by any of the other studies examining vaccine distribution preferences.”

Additions to: Discussion (later part)

“Moreover, our finding regarding prioritising candidates according to employment status, and particularly potential income losses, may also indirectly point to a pronounced public support for vaccine donations to the Global South: Employment security is much lower in low- and middle-income countries, especially because of the high share of workers in the informal sector where substantial income losses are more likely to occur during a lockdown or pandemic-induced economic downturns.”

4. The temporal nature of the public views at various stages of the pandemic and vaccination campaigns should also be noted.

Thank you very much for this important comment. We agree that there is an important temporal element to public opinion during the different stages of the pandemic and that this should be acknowledged accordingly when reflecting on our findings. We have now included an additional paragraph in the Discussion section of the paper, where we had already stated a somewhat related point, i.e. the relevance of the pandemic trajectory over time across sampled countries.

Additions to: Discussion

“A number of limitations are worth noting. First, in addition to the cross-country differences in COVID-19 infection and vaccination rates discussed above, participants’ preferences may partly also be driven by cross-country variation in the pandemic trajectory over time. Given that our study utilises only cross-sectional survey data, we are not able to empirically examine any changes in public perceptions regarding the pandemic over time. Yet, the pronounced temporal nature of the pandemic needs to be kept in mind when reflecting on our findings. This includes, for instance, (i) notions of scarcity in the early phases of the vaccination campaigns, which may have affected perceptions about their value, (ii) potential erosion of public trust in COVID-19 vaccines during the suspension of the AstraZeneca/Johnson and Johnson vaccines, and (iii) increases in conspiracy beliefs related to vaccination through increasing spread of misinformation via social media channels (Steinert et al., 2022). More generally, however, citizens’ overall assessment of the pandemic situation likely varies strongly with the various stages of the pandemic, which we are at least partly able to capture by exploiting the different timing of our survey launches and thus some country (and regional) differences in infection rates at the time of data collection.”

5. During the analysis of the main results (Table 2), were respondents' characteristics, including being in a high-risk group, controlled for? It will be helpful to clarify this in the paper. Yet even with the inclusion of such controls, it is likely that the characteristics of the sample from Germany and the timing of the fieldwork during which Germany had high infection rates while being behind the rest of the EU countries in vaccination rates play a role in the study findings from Germany. A stark difference between country samples e.g. is the high rate of those respondents who were in the high-risk group in Germany. This factor, even after being controlled for, very much likely has influenced respondents' attitudes towards sharing the vaccines with the rest of the world.

Thank you very much for highlighting these important points. We are utilising a conditional logit model, which includes respondent-fixed effects, and are moreover clustering standard errors at the respondent level. Thus, respondents’ socioeconomic characteristics, such as their COVID-19 risk status, are always controlled for, but are absorbed by the fixed effects and, thus, not measurable. Since respondents’ characteristics do not vary across the eight choice sets of the DCE, it is not possible to ‘control’ for their impact in the traditional sense because there is no variance to exploit within respondents across choice sets. The attributes of the candidates in the DCE, however – whose impact we are mainly interested in – do vary across choice sets. In view of this, a conditional logit with fixed effects at the respondent level seemed to be the most robust approach and is commonly used in the DCE literature. However, we do take a closer look at the potential impacts of respondents’ socioeconomic characteristics by means of our heterogeneity analysis. Here, the interaction of respondents’ (fixed/time-invariant) characteristics with the (varying) attributes of the candidates in the DCE allows us to measure the impact of respondents’ characteristics (such as age, sex, education level, own risk status) on the probability of choosing a candidate from the Global South (i.e. due to the interaction, their effect is not absorbed by the fixed-effects model).

Nevertheless, we agree with your concern that the high proportion of high-risk respondents in the German sample probably played a role for the differential findings revealed for Germany. As we stated in our response to comment (1), the discrepancies in the findings from Germany compared to those from the other countries most likely resulted from the different composition of the German sample on the one hand (though not only in terms of high-risk respondents, but also in terms of older and lower educated respondents), and the high infection rates and low vaccination rates during the data collection in Germany on the other hand. Regarding this point, we would like to kindly ask you to refer to our detailed response and additional analyses outlined in relation to comment (1) above. Given that this point about the sample composition, vaccination and infection rates was raised both in comments (1) and (5), we will here only list the additional changes not yet addressed in our response to comment (1).

To improve clarity and transparency regarding the utilised model and the way in which it accounts for respondents’ characteristics, we have included additional explanations in the methods section:

Additions to: Materials and Methods (Subsection: Statistical Analyses):

“The empirical analysis comprised two steps. First, we estimated the main effects model for each country separately to assess the impact of the four candidate attributes (country of residence, age, COVID-19 mortality risk, employment status) on the probability of choosing a specific candidate. The statistical model we utilised was a conditional logit model (with respondent fixed-effects) with standard errors clustered at the respondent level. Thus, in the main effects model, we regressed respondents’ vaccine allocation choice on the attribute levels of the respective candidate in each scenario. The potential impact of respondents’ own characteristics was controlled for and absorbed by the fixed effects, given that respondents’ characteristics do not vary across the eight choice sets of the DCE.

Second, we examined heterogeneity in the effect of the country of residence attribute by adding interaction terms between the country of residence and heterogeneity variables to the above regression. Thus, the interaction of the respective heterogeneity variable (e.g. sex or age category) with the country of residence attribute – which varies across choice sets – allows us to examine the importance of respondents’ characteristics while still using a fixed-effects model. A pre-analysis plan along with justifications of any deviations thereof is accessible via https://osf.io/72jrq/.”

Moreover, the notes accompanying all tables and figures that present regression outputs have been extended by the following small addition:

“Coefficients are odds ratios based on conditional logit estimations (respondent-level fixed effects) with standard errors clustered at the respondent level.”

6. It will be interesting to see a chart using the pooled data depicting the ratio of responses to the 8 choice sets of the DCE (as shown in Table 4).

Thank you for bringing this to our attention. We agree that such a chart is indeed highly informative and useful as an initial presentation of the DCE results. We have now extended Table 4 by adding seven more columns containing the ratio of responses to the eight choice sets of the DCE, both for the pooled sample and for each country individually. We have moreover moved the table from the Methods section to the main manuscript (thus, now it is Table 2). These additions to the table show how respondents decided to distribute the vaccine in each of the eight choice sets and allow us to identify in a descriptive manner where and how, for example, German respondents decided differently than Spanish respondents. The observed proportions confirm the same patterns identified in the main effects model of the empirical analysis (now Table 3) and moreover reveal in which choice sets country differences were particularly dominant (e.g. choice sets 2, 3, 5, 7 and 8 in the case of Germany vs the other countries).